# SEM Evaluation of the Marginal Accuracy of Zirconia, Lithium Disilicate, and Composite Single Crowns Created by CAD/CAM Method: Comparative Analysis of Different Materials

**DOI:** 10.3390/ma16062413

**Published:** 2023-03-17

**Authors:** Francesco Ferrini, Gaetano Paolone, Giovanna Laura Di Domenico, Nicolò Pagani, Enrico Felice Gherlone

**Affiliations:** 1Dental School, IRCCS San Raffaele Hospital, Vita-Salute San Raffaele University, 20132 Milan, Italy; 2Oral Surgery School, Vita-Salute San Raffaele University, 20132 Milan, Italy

**Keywords:** CAD/CAM, composite, digital impression, lithium disilicate, marginal accuracy, scanner, scanning electron microscope, zirconia

## Abstract

(1) Background: The objective of this in vitro study is to evaluate the marginal accuracy of crowns created by CAD/CAM. (2) Methods: A customized chrome-cobalt (Cr-Co) implant abutment simulating a maxillary right first molar was fixed in a hemi-maxillary stone model and scanned. In total, 27 crowns were fabricated, including 9 lithium disilicate crowns, 9 composite crowns, and 9 zirconia crowns. The measurements were determined by scanning electron microscopy. Descriptive analysis was performed using the mean and standard deviation, while the Kruskal–Wallis test was performed to determine whether the marginal discrepancies were significantly different between each group (*p* < 0.05). (3) Results: The lowest marginal gap value was reported for zirconia (21.45 ± 12.58 µm), followed by composite (44.7 ± 24.96 µm) and lithium disilicate (62.28 ± 51.8 µm). The Kruskal–Wallis tests revealed a statistically significant difference (*p*-value < 0.05) in the mean marginal gaps between different materials. (4) Conclusions: The proposed digital workflow can be a viable alternative for fixed prosthetic rehabilitations. The best performance in terms of marginal gap was achieved by zirconia crowns, but all three materials demonstrate marginal closure below the clinically accepted threshold value (120 µm). Clinical significance: although significant differences were reported, the investigated CAD/CAM materials showed clinically acceptable marginal gaps.

## 1. Introduction

There are four specific attributes that a dental restoration must possess in order to be considered successful, which are: marginal adaptation, biocompatibility, aesthetics, and mechanical strength [1]. Research, innovation, and new technologies have moved dentistry toward an all-digital flow by replacing classical analog techniques [2]. Digital techniques allow clinicians to save time and material costs, as well as provide a comfort advantage for the patient by not having to take impressions with analog materials [3]. The traditional workflow requires more time and manual skills, since it begins with a conventional impression, which is then transformed into plaster, on which the technician builds the prosthesis [4]. In the digital workflow, a technician uses the 3D images generated from the optical impression, obtained by the use of an intra-oral scanner, to manipulate a virtual restoration and mill the restoration from a digital file. The evolution of computer-aided design and computer-aided manufacturing (CAD-CAM) technology has, in fact, been remarkable, allowing for the creation of dental restorations through a production process that involves direct or indirect digitalization. This process permits the use of computer technology to design, analyze, and either add or subtract restorations [5]. Digital technologies also enabled the development of high-performance materials, such as hybrid polymers and ceramic CAD/CAM materials. Among these, zirconia, lithium disilicate, and composite are the most widely used materials in the digital workflow. In the last few years, they have undergone important mechanical and physical changes, increasing their overall performance [6].

Marginal fit is an important factor for the success of tooth restoration. Marginal discrepancies in a dental restoration can have several negative consequences, such as increased plaque accumulation, altered microflora distribution, and a higher risk of caries in the abutment teeth in the case of natural teeth [7,8,9]. There is currently no agreement on the maximum marginal discrepancy that is clinically acceptable, as there is no clear consensus based on the scientific evidence available. Different studies reported that a 50–120 mm gap is clinically acceptable, while other studies suggested gaps of less than 100 mm [10,11,12]. In their study, Holmes et al. [13] defined the marginal gap (MG) as the distance between the cervical margin of the casting and the preparation margin. They also defined the absolute marginal discrepancy (AMD) as the angular combination of the MG and the extension error (either overextension or underextension), measured from the cavosurface of the preparation to the cervical margin of the cast. The MG is particularly significant since it is the cemented surface that is exposed to the oral environment and may dissolve, leading to microleakage. In relation to this, according to the American Dental Association Specification No. 8, luting agents should have a film thickness of 25–40 mm [14]. On the other hand, the AMD is crucial because it affects plaque accumulation by indicating the extension of crown margins relative to preparation margins.

McLean and Von Fraunhofer, in an in vivo study on 1000 restorations with 5-year follow-ups, concluded that the degree of acceptable marginal discrepancy is up to 120 microns [15]. Since marginal fit is a fundamental factor for assessing the quality of restorations, many studies, including literature reviews and systematic reviews, have addressed this issue [16,17,18,19,20,21,22]. The results reported different values ranging from 5.49 µm to 280 µm, mainly due to differences, in the study designs, i.e., measurement method, sample size, quantity of measurements, restoration material, type of microscope, type of abutment, and finish line [16,17,23]. Due to the involvement of various materials and multiple stages in both clinical and laboratory settings, it is in fact inevitable to encounter marginal inaccuracies during the conventional fabrication process. Boitelle et al. [24] reported that a reliable CAD-CAM system should be able to deal with different materials and machine them appropriately and precisely for the production of high-quality restorations. Misfits in CAD-CAM restorations can arise due to constraints in the design of restorations with software programs, hardware limitations related to 3D image capturing, and the milling process [25,26]. Nevertheless, the range of restorative material properties has been reported to influence the marginal adaptation of CAD-CAM restorations [27,28,29,30,31]. Few studies and conflicting results are available in the literature relating to the marginal fit of monolithic CAD/CAM materials. The aim of the present study was therefore, to compare the marginal gap values of prosthetic crowns of different monolithic materials, produced by the same CAD/CAM method (CEREC, Dentsply Sirona). The null hypothesis tested was: there is no difference in marginal adaptation among the different groups of materials produced with the same production and impression methods. 

## 2. Materials and Methods

The dental laboratory prepared a plaster model, which was a right hemi-maxilla, in order to simulate the situation of a dental-supported single crown in position #16.

A customized cobalt-chrome (Cr-Co) abutment was manufactured according to the STL file generated by a 3D CAD modeling software (Rhinoceros 5; Robert McNeel & Associates, Seattle, Washington, USA). The abutment provided a diameter of 1 cm, height of 1 cm, a taper of 3°, and a 2-mm-deep 360° chamfer as a marginal design. (Figure 1a) The finish line level was assumed to be similar to that of a natural tooth, i.e., it was more apical in the buccal and palatal areas compared to the interproximal regions, which were more coronal. Two vertical slots were created on the base of the abutment, as reference points, in order to allow the correct repositioning of the abutment in case of removal. Sandblasting was performed to improve surface roughness and facilitate the scanning process. Then, the abutment was mounted and fixed in a hemi-maxillary stone model by using cyanoacrylate. (Figure 1b) The finish line was kept at the gingival level with respect to adjacent teeth. The model was reduced in correspondence with the Co-Cr abutment to mimic the presence of gingival sulcus. A hole at the model bottom was created for easy abutment removal. 

A scan of the hemi-model was created with Cerec Primescan (Dentsply Sirona, Charlotte, NC, USA) and ended with the exportation of data in Standard Tessellation Language (STL) format. After scanning and creating the model, the finish line was marked, and the crown was designed on a Cerec Primescan touch desktop. (Figure 2) The core displayed a uniform 1 mm thickness thanks to the coping offset function, and the luting space was adjusted to 30 μm.

Before starting the coping production, a test coping was produced in order to evaluate the right fit according to abutment geometry and prosthetic materials. Once the ideal fit was identified, the same parameters were established and kept constant throughout the entire process. The design was evaluated for proper contacts and contours, and the crowns were fabricated in succession with the Cerec inLab MC XL (Dentsply Sirona, Charlotte, CN, USA) milling machine and sintered (for G1 and G2) with the Cerec SpeedFire (Dentsply Sirona, Charlotte, CN, USA) following the manufacturer’s recommendations. The time for each crown production was recorded.

In this study, 3 materials (Table 1) were used for a total of 27 restorations (*n* = 9) (Figure 3):-Group 1 (*n* = 9): zirconia (Katana Zirconia STML);-Group 2 (*n* = 9): lithium disilicate (Cerec Tessera);-Group 3 (*n* = 9): composite (Katana Avencia Block).

According to the restoration’s material and the manufacturer’s indications, different burs were used for the milling process: Shaper 25 RZ and Finisher 10 burs were used for zirconia, while Cylinder Pointed Bur 12S (Dentsply Sirona, Charlotte, CN, USA) and Step Bur 12S (Dentsply Sirona, Charlotte, CN, USA) were used for lithium disilicate and composite. Before beginning the scanning electron microscope (SEM) investigation, a fit check was performed for all the specimens. A sample size of 15,000 points with a tolerance of 0.001 mm was used in the 3D analysis software (Geomagic Control 2022, 3D Systems, Rock Hill, SC, USA) to check differences among the produced crowns. Then, each crown was positioned on the abutment and kept in place by a hand-crafted elastic metal spring capable of retaining the crown in the ideal position and at the maximum depth of fit. No material was inserted at the interface abutment-restoration.

Observations were made with a PhenomPro X SEM (Thermo Fisher Scientific Inc.; Waltham, MA, USA) at Metalcoating SRL (San Giuliano Milanese, Milan, Italy). By positioning the stump on specially designed metal frames, it was possible to keep the various specimens in the same position throughout the various scans with a constant inclination of 25° to the perpendicular electron beam. This ensured that the specimen’s inclination and, consequently, the angle of incidence of the scanning electron beam were always the same. Photomicrographs (Figure 4) were obtained from the center of each sample, with magnification up to 580× and then assessed by a calibrated and blind examiner.

The marginal gap at the restoration-abutment interface was measured at the vestibular (V), palatal (P), mesial (M), and distal (D) aspects. For each side, the point at minimum radial distance between the circumference and the corresponding aspect was identified, resulting in the V, P, M, and D points (Figure 5). Then, additional points on the circumference were identified at 25% and 75% of angular distance between two consecutive points among the ones previously identified.

Descriptive statistics were presented as means and standard deviations. The Kruskal–Wallis and the Dunn tests with the Bonferroni correction were used to analyze differences among groups. The statistical analyses were performed using the software package (STATA 14.1; StataCorp LLC, Lakeway Drive, TX, USA).

Figure 6 shows the experimental design workflow.

## 3. Results

The SEM analysis of marginal gap performed on 324 points (12 for each coping) showed a total mean value of 21.45 ± 12.58 μm for zirconia, 62.28 ± 51.8 μm for disilicate, and 44.7 ± 24.96 μm for composite crowns, with statistically significant differences between the three groups (Table 2). The sample distribution is represented by the boxplot in Figure 7.

When considering all the measurements, pairwise comparisons using Dunn’s test indicated that group 1 scores were observed to be significantly lower than those of group 2 (*p* < 0.0001) and group 3 (*p* < 0.0001). No other differences were statistically significant.

Results varied when the gap’s location was taken into account (Figure 8).

In terms of vestibular gaps, group 1 scores were significantly lower than those of group 2 (*p* < 0.0001) and group 3 (*p* < 0.0001), and there was a statistically significant difference between group 2 and 3 (*p* = 0.0232). 

Regarding the palatal gap, group 3 scores were significantly different from those of group 1 (*p* = 0.0014) and group 2 (*p* < 0.0001).

At mesial aspect, the post-hoc test was significant for group 1 vs. group 2 (*p* < 0.0001) and group 1 vs. group 3 (*p* = 0.0001), while in the distal aspect, it was only significant for group 1 vs. group 2 (*p* = 0.0047).

The total milling time was 7 h and 20 min (for all the 27 crowns), with an average of 16 min per crown. The material with the longest production time was zirconia (Table 3).

## 4. Discussion

An increasing demand for esthetic dental restorations has led to the popularity of ceramic restoration systems. Contemporary esthetic materials for indirect restorations are widely used in dentistry as they meet the functional and esthetic demands of both the anterior and posterior regions of the mouth. The long-term success of indirect restorations relies heavily on the marginal and internal fit. When there is a misfit, plaque can accumulate, potentially causing caries or periodontal diseases, especially in restorations with subgingival margins [9]. Moreover, an inadequate fit can lead to cement dissolution, allowing bacteria to infiltrate and decreasing fracture resistance [16]. In computer-aided design and computer-aided manufacturing (CAD-CAM) restorations, a not-exact fit could result from: (1) limitations in designing restorations with a software program; (2) hardware limitations regarding 3D image capturing; (3) characteristics of the milling process [25,26].

The present study examined the marginal fit of three different CAD/CAM materials that are widely clinically used. Since significant differences were observed among the investigated materials, the null hypothesis has to be rejected. It was therefore, concluded that the performance of a specific CAD/CAM system, in relation to marginal adaptation, is influenced by the type of restorative material. Marginal adaptation is a very important clinical parameter because a wide marginal gap can determine postoperative sensitivity, secondary decay, discoloration of the margins, an unesthetic appearance, and mechanical failure. A recent literature review reported that the absence of consensus regarding the precision of fit of different crown materials should be related to differences in experimental protocols [17].

The final fit of the restoration can, in fact, be affected by several factors. Among them, the design of the finishing line is one of the most investigated factors, with some studies reporting no significant differences between horizontal and vertical finish lines [32,33,34,35], while others reporting that shoulder results in a more accurate one [36,37,38,39]. In our study, we opted for the latter, choosing a 360° deep chamfer. Among the other investigated factors influencing marginal adaptation are the acquisition and fabrication workflow and the choice of restorative material. Up to now, various materials with different compositions, physical and optical properties have become available for CAD/CAM workflow [40,41], nevertheless, investigations on their effects on restoration’s marginal adaptation have reported conflicting findings [42,43]. 

In the present in vitro investigation, although significantly different, all the tested crown materials showed clinically acceptable values of marginal discrepancy. The size of the marginal gap of all milled crowns in the present study was in fact lower than 120 μm, which has been identified as the threshold value for a clinically acceptable marginal gap allowing to achieve a good long-term success of prosthetic crowns [44,45]. These results are in agreement with those achieved in similar previous studies [46,47,48,49].

Particularly, the zirconia crowns showed the best precision of fit at the preparation margins (21.45 ± 12.58 μm). This could be explained by several factors, such as the fact that dental CAD/CAM systems were developed for processing polycrystalline materials, which led to the creation of more accurate results [30,47,48,49]. Furthermore, the investigated materials were grinded using the manufacturer’s suggested bur system, and this characteristic may have accounted for a better marginal fit.

In the present study, we focused on the marginal gap as it is considered a reliable analysis for crown adaptation assessment [26,50]. While in vitro testing offers a standardized approach to preparation design, technique, and experimental performance, caution should be exercised in interpreting the results as they may not fully replicate the clinical setting. Nevertheless, the use of standardized conditions facilitates more reproducible assessments.

This study investigated three types of materials: zirconia, composites, and lithium disilicate.

Each of these materials has its own unique properties and characteristics, which make them suitable for different clinical situations.

Zirconia is a type of ceramic material that is known for its strength and durability. It is highly resistant to wear and fracture, making it an excellent choice for restorations in areas of the mouth that are subject to high stress, such as molars. Additionally, zirconia is biocompatible, which means that it is unlikely to cause an adverse reaction in the body. Zirconia is also highly aesthetic and can be matched to the color of natural teeth. Composite, on the other hand, is a tooth-colored resin material that is used for dental restorations. It is a popular choice for fillings and bonding procedures because it can be easily molded and shaped to match the contours of the teeth. Composite is also highly esthetic and can be color-matched to the surrounding teeth for a natural-looking result. However, it is not as strong as zirconia and may not be suitable for restorations in areas of the mouth that are subject to high stress. Lithium disilicate is another type of ceramic material that is used for dental restorations. It is known for its strength and esthetic qualities, making it a popular choice for crowns, veneers, and other types of restorations. Lithium disilicate is also biocompatible and can be matched to the color of natural teeth. However, similar to composite, it may not be as strong as zirconia and may not be suitable for restorations in areas of the mouth that are subject to high stress.

Of the monolithic CAD/CAM materials that have been recently introduced for cuspal coverage, indirect restorations on severely damaged teeth, zirconia has undergone significant advancements. High-translucency zirconia has emerged as a promising replacement for tetragonal zirconia, particularly for monolithic single-tooth restorations. By introducing a variable amount of cubic phase, which is optically isotropic, the translucency of the material has been enhanced. However, this comes at the cost of strength and toughness due to the lack of transformation toughening and the coarser microstructure [51]. Initially, high translucency zirconia was considered less suitable for posterior restorations and was recommended only for anterior applications because cubic grains are wider than tetragonal ones, generating more stabilizing oxides and making the tetragonal phase more susceptible to aging [52]. However, recent studies have led to the development of various types of zirconia with different percentages of cubic phase, ultimately creating zirconia-based materials specifically designed for the posterior sectors, with a suitable balance between optical and mechanical properties [53].

The use of CAD/CAM composite blocks, also known as nano-hybrid ceramic, resin nano-ceramic, resin ceramic, nanohybrid restorative materials, resin-based composite blocks, or hybrid composites, has become increasingly popular due to their superior quality compared to conventional composite materials used in conventional analogic dental procedures for both direct and indirect restorations [54]. This is attributed to the standardized industrial production processes involving high temperature and/or high-pressure polymerization, which facilitate maximum polymer cross-linking and enhance the material’s properties [55]. CCBs offer additional advantages, such as easy finalization without the need for firing after milling, simple finishing and polishing, and repairability compared to ceramics [55].

In our study, Katana Avencia, a composite block was analyzed. This study analyzed marginal fit, nevertheless, it has to be reminded that mechanical properties shall be always considered, being CCBs affected by monomer compositions, especially if specific types of monomers are contained (e.g., TEGDMA) [56,57,58,59]. 

IPS e.max (Ivoclar Vivadent, Schaan, Liechtenstein) was introduced in 2005 and can be utilized through two techniques: the lost wax hot press method (IPS e.max Press) or the digital CAD/CAM approach (IPS e.max CAD) [60,61]. IPS e.max Press and IPS e.max CAD have a crystal content of 70% and exhibit similar mechanical characteristics, such as the modulus of elasticity, thermal expansion coefficient, and chemical solubility. The only notable difference is in their flexural strength, which is 360 MPa for IPS e.max CAD and 400 Mpa for IPS e.max Press, reflecting the difference in crystal size [60]. Previous research comparing the marginal fit of lithium disilicate crowns created via both techniques revealed that pressed restorations displayed better marginal fit [62]. However, recent studies have suggested that CAD/CAM-fabricated crowns exhibit smaller gaps, typically ranging from 33 to 77 μm [63,64]. Our study supports these findings, as we recorded an average marginal gap of 62 μm. These discrepant results may be attributable to advancements in CAD-CAM technology, particularly in digital planning and fabrication parameters, which have improved the precision of the restoration process.

Several marginal gaps for CAD/CAM lithium disilicate were in fact reported in the literature: 33.30 μm [65], 84 μm [62]; the latter was obtained with an older CAD/CAM system (E4D milling engine) in comparison to the first one. Material is not the only factor influencing the adaptation and marginal fit of CAD/CAM-milled crowns. Hamza et al. [31] reported significantly different marginal fits for zirconia or lithium disilicate if different CAD/CAM systems were used (Cerec inLab, Kavo Everest). If the Everest CAD/CAM system (Everest Bio ZS) was employed, the authors reported better values (lower marginal gaps) for zirconia blocks (Everest Bio ZS blocks) than lithium disilicate ones (E.max CAD blocks for Everest). Opposite results were encountered if the Cerec inLab system was used. The authors reported that differences could be attributed to the scanning process and the number of axes on the milling machines of the two systems: Everest Energy has a 5-axis unit, while the Cerec inLab MC XL has a 3-axis milling unit. Weather opposite results for both materials were obtained for the two CAD/CAM system, was not discussed by the authors. In our study, the same machine was used (Cerec inLab MC XL) but opposite results (between zirconia and lithium disilicate) were reported. Differences may be due to the type of burs (not specified) in the milling unit. The biggest difference is that both studies are investigating lithium disilicate, but in the present study we analyzed Cerec Tessera (Dentsply Sirona), while Hamza et al. investigated Ivoclar’s lithium disilicate (e.Max). Other studies confirmed that marginal adaptation depends on the CAD/CAM system [66,67,68,69,70]. It is reported by the authors investigating different systems that discrepancies are due to the limitations of the CAD/CAD systems and to the capability of the operators to customize their specific parameters.

Different measuring methods of marginal discrepancies have been described in the literature, including the direct-view technique by using scanning electron microscopy [32,71], or optical microscope [32,71,72,73,74]; the cross-sectioning technique [16,75]; radiographic techniques [76]; the impression replica technique [77,78]; profilometry [79]; digital quantitative evaluation [80,81]; and microcomputed X-ray tomography (mCT) [82,83,84,85].

In our study, the measurement of the marginal gap was performed using the direct view technique with a SEM. In order to ensure accuracy and repeatability of the measurements, custom support frames were positioned at a constant angle of 25° to the incident electron beam. Moreover, precise positioning of the specimens was also aided by the reproducible and repeatable anatomy of the crowns, since they were all milled from the same initial design.

Furthermore, the selection of measurement points is another crucial point. Studies have recommended that at least 10 points be randomly selected to measure the adaptation of the restoration and that measurements should be performed throughout the restoration. [29] In this study, 12 reference points were identified and measured for all the tested specimens. The process of crystallization can potentially affect the marginal fit. To investigate this, Gold et al. [86] assessed the marginal fit of CAD-CAM-fabricated lithium disilicate crowns both before and after the crystallization process. Their analysis revealed a significant difference in marginal discrepancy, with measurements of 42.9 mm prior to crystallization and 57.2 mm after.

It has to be also highlighted the importance of relating the findings of this article to other mechanical tests, in particular the ones related to wear. The wear mechanism in dental materials can be categorized into two types: two-body and three-body wear. Two-body wear involves the contact between two surfaces, where the opposing surfaces rub against each other. This type of wear is commonly seen in materials such as ceramics, where a hard surface is in direct contact with another hard surface. In two-body wear, the wear rate is typically lower, and the wear patterns are predictable. The wear mechanism can be influenced by factors such as load, surface roughness, and material properties [87,88].

On the other hand, three-body wear occurs when particles or debris become trapped between the two surfaces in contact. The debris acts as an abrasive and leads to faster wear rates and more unpredictable wear patterns. Three-body wear is common in materials such as composites and dental amalgam, where particles can be released during chewing or abrasive forces. The wear rate can be influenced by factors such as particle size and shape, volume fraction of particles, and the properties of the matrix material. Therefore, an understanding of the wear mechanisms in dental materials is crucial for the development of durable and long-lasting restorations [87,88].

Relating to the novelty of the current study, we outlined the better marginal fit of zirconia frameworks. A marginal seal is the interface between the crown and the tooth that is created to prevent bacteria and debris from entering the space between the crown and the tooth. This seal is essential for the long-term success of a dental restoration. Zirconia is a ceramic material that has become popular for use in dental crowns due to its superior strength and aesthetics. One of the key advantages of zirconia crowns over other ceramic materials is their ability to create a better marginal seal.

Zirconia crowns are known for their excellent fit and accuracy [46], which is due to their ability to be precisely milled using computer-aided design and manufacturing (CAD/CAM) technology. This precise fit helps to create a better marginal seal, which reduces the risk of microleakage and subsequent decay. Zirconia is also a biocompatible material, which means that it is less likely to cause an allergic or inflammatory response in the surrounding tissues, which can also help to promote better marginal sealing.

Furthermore, zirconia crowns have a high resistance to wear and fracture, which means that they are less likely to break down over time, thereby maintaining the marginal seal for a longer period [46]. In contrast, other ceramic materials, such as porcelain, have a tendency to chip or wear down, which can compromise the marginal seal and increase the risk of decay. Overall, the superior marginal sealing properties of zirconia crowns make them an excellent choice for dental restorations that require high strength, accuracy, and longevity.

This study has some limitations related to study design. As an in vitro experiment, this could not include in vivo factors, such as patient cooperation or contamination with biologic fluid during the scanning process. It must indeed be emphasized that while fit is a critical factor in the clinical success of complete crowns, it is only one of several contributing factors. Another limitation is that only one scanner system and one milling machine was used. Further study and clinical applications should be performed involving cyclic fatigue [89] in the experimental design to better simulate clinical scenarios. Furthermore, further studies on the biocompatibility, biomechanics, and biomimetic properties [90,91] of the investigated materials should be performed. 

## 5. Conclusions

Taking into account the limitations of the present in vitro investigation, the researchers were able to draw two important conclusions. Firstly, all the tested crown types showed marginal gaps that fell within the acceptable clinical range. This is a significant finding, as marginal adaptation is essential for preventing bacterial invasion and decay of the underlying tooth structure. Secondly, the study found that zirconia crowns exhibited better marginal adaptation than lithium disilicate and composite crowns. However, it is important to note that the results of this study should be interpreted with caution, given the in vitro setting and the need for further clinical trials to validate the findings. Additionally, the study was conducted under controlled laboratory conditions and may not necessarily reflect clinical reality. Therefore, these findings should be taken as a starting point for further research in this area.

## Figures and Tables

**Figure 1 materials-16-02413-f001:**
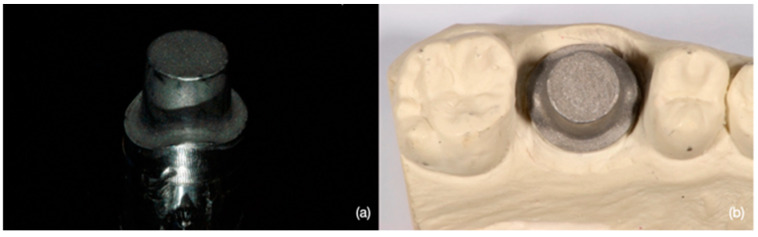
(**a**) The customized Cr-CO abutment generated by a 3D CAD modeling software. (**b**) The Cr-Co abutment is positioned in the model.

**Figure 2 materials-16-02413-f002:**
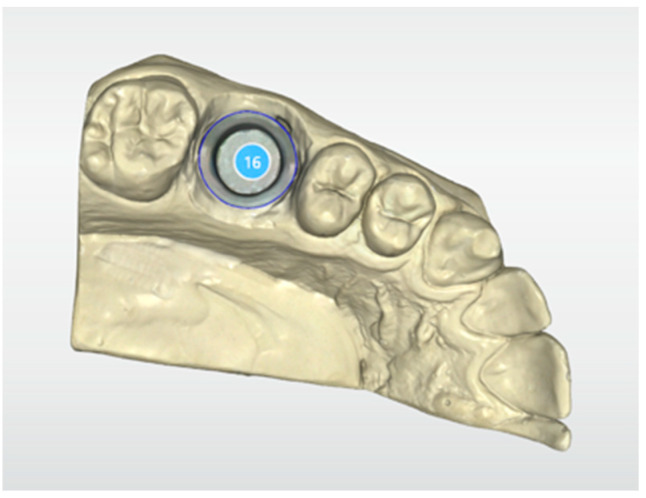
Occlusal view of the 3D model obtained with Cerec Primescan. 16: first uppor right molar.

**Figure 3 materials-16-02413-f003:**
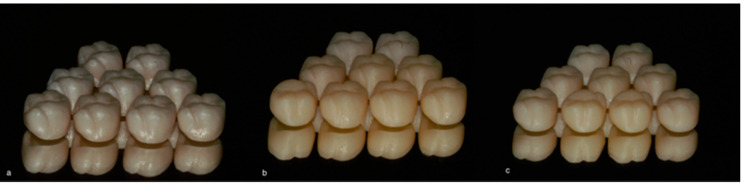
Zirconia (**a**), composite (**b**), and lithium disilicate (**c**) crowns.

**Figure 4 materials-16-02413-f004:**
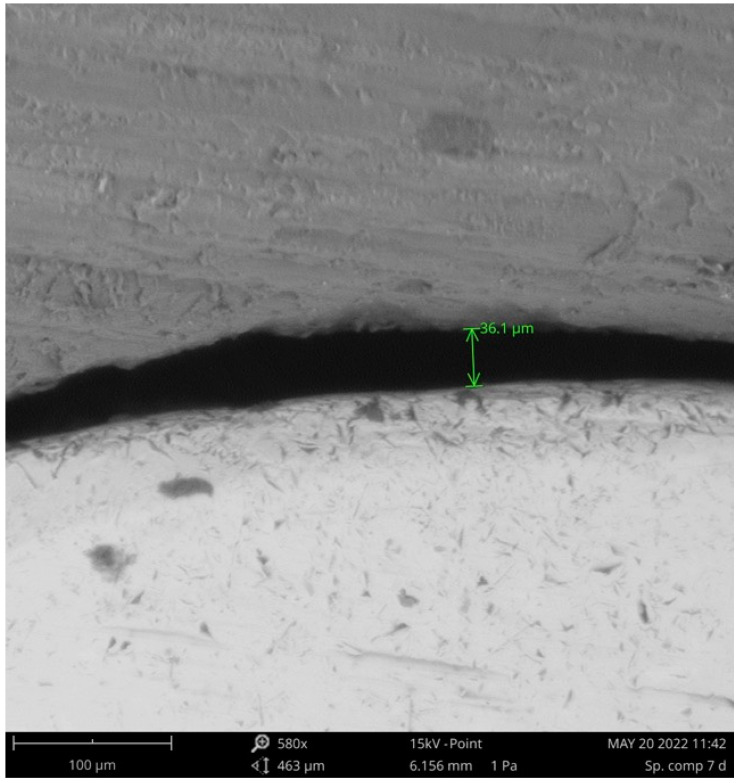
A 580× SEM image for evaluation of marginal gap at restoration-abutment interface.

**Figure 5 materials-16-02413-f005:**
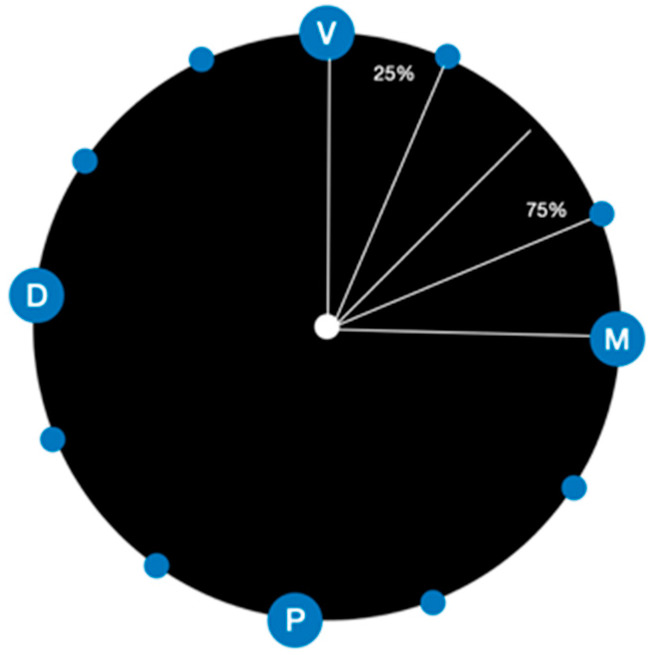
Scheme of reference points.

**Figure 6 materials-16-02413-f006:**
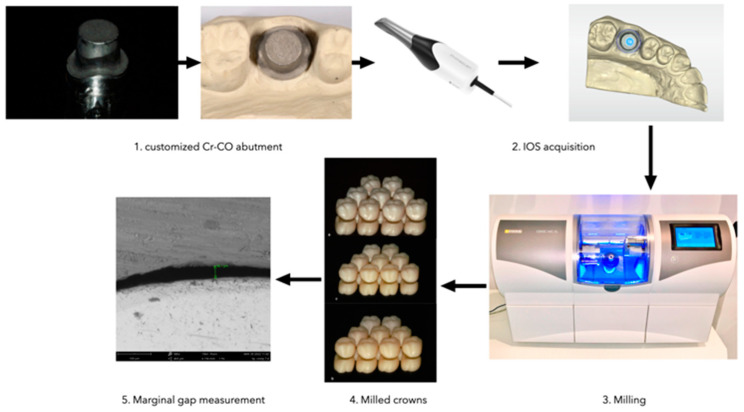
Schematic representation of the experimental design workflow.

**Figure 7 materials-16-02413-f007:**
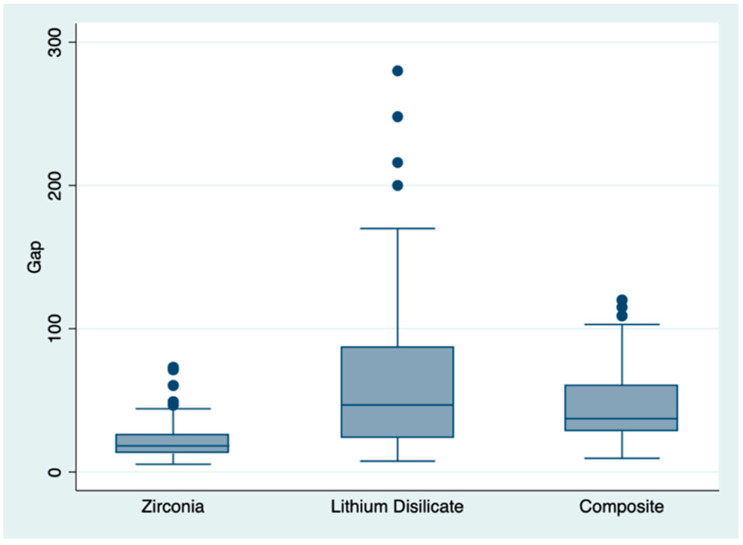
Boxplot representing the sample distribution. Outlier values are indicated by “●” symbols.

**Figure 8 materials-16-02413-f008:**
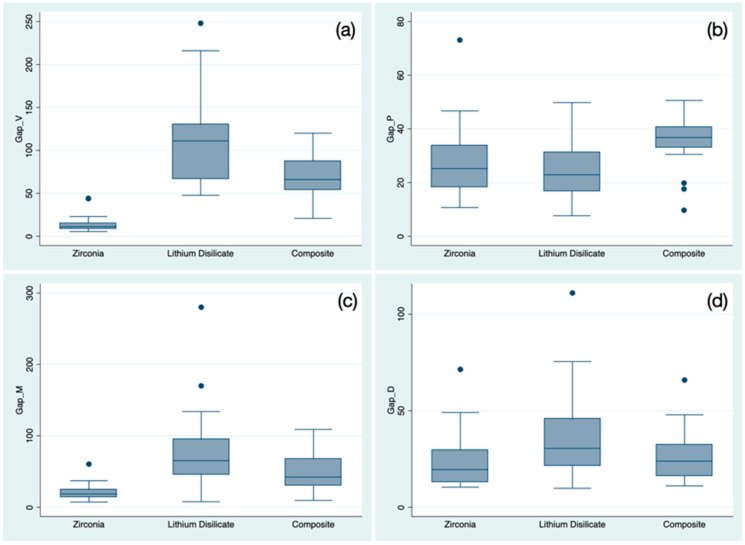
Boxplot representing the sample distribution according to the gap’s location: (**a**) vestibular gap; (**b**) palatal gap; (**c**) mesial gap; and (**d**) distal gap. Outlier values are indicated by “●” symbols.

**Table 1 materials-16-02413-t001:** Composition of the investigated materials.

Product Name	Type	Manufacturer	Composition	Lot. Number
Katana Zirconia STML	Zirconia	Kuraray-Noritake, Miyoshi, Japan	5–5.5 mol% yttria-containing zirconia	EERQF
Cerec Tessera	Lithium Disilicate	Dentsply Sirona, Charlotte, CN, USA	Li_2_O_5_Si_2_: 90% 16,007,942 Li_3_PO_4_: 5% LiAlSi_2_O_6_ (Virgilite): 5%	16007942
Katana Avencia Block	Composite	Kuraray-Noritake, Miyoshi, Japan	Matrix: UDMA, TEGDMA; filler 62 wt%; compressed nanosized fillers: aluminum oxide (20 nm), SiO_2_ (40 nm)	000123

**Table 2 materials-16-02413-t002:** Mean and standard deviation (μm) of the gap between restoration and abutment.

	Group 1 (Zirconia)	Group 2 (Lithium Disilicate)	Group 3 (Composite)	*p*-Value ^a^
Vestibular (*n* = 27)	14.24 ± 9.7	111.64 ± 48.94	69.72 ± 25.8	0.0001 *
Palatal (*n* = 27)	27.51 ± 12.81	23.96 ± 9.43	35.61 ± 8.92	0.0001 *
Mesial (*n* = 27)	20.68 ± 10.24	77.3 ± 55.81	47.77 ± 23.67	0.0001 *
Distal (*n* = 27)	23.41 ± 13.85	36.22 ± 21.33	25.71 ± 11.79	0.0112 *
Total (*n* = 108)	21.45 ± 12.58	62.28 ± 51.8	44.7 ± 24.96	0.0001 *

^a^ Kruskal–Wallis test. * Statistically significant.

**Table 3 materials-16-02413-t003:** Time of crown production.

	Group 1 (Zirconia) (*n* = 9)	Group 2 (Lithium Disilicate) (*n* = 9)	Group 3 (Composite) (*n* = 9)
Total milling time	3 h 30 min	2 h 59 min	1 h 43 min
Milling time/crown	23.46 min	19.92 min	11.44 min
Total sintering time	4 h 39 min	NA	NA
Sintering time/crown	31 min	NA	NA

## Data Availability

The data presented in this study are available upon request from the corresponding author.

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
