# Peer review of "SEM Evaluation of the Marginal Accuracy of Zirconia, Lithium Disilicate, and Composite Single Crowns Created by CAD/CAM Method: Comparative Analysis of Different Materials"

_materials, 2023, doi:10.3390/ma16062413_

Round 1
Reviewer 1 Report
The article seems to be quite relevant in the field of biomaterial based on dental Composite resin that SEM evaluation of the marginal accuracy of zirconia, lithium 2 disilicate, and composite single crowns made by CAD/CAM 3 method: comparative analysis of different materials. It is very good that there is a original article on the mechanical and aesthetic behavior of dental materials in the literature. However, the article can be made even better by making the following corrections
è Introduction
1) How does the organic structure of TEGDMA affect the surface roughness of the composite material? The authors should provide more literature information on this subject. Because Surface roughness is an important parameter in terms of aesthetic and mechanical behavior. For example, surface roughness behavior can affect wear mechanisms and adhesion mechanisms. For this reason, authors should improve the in title section of the article. In addition, the results obtained by the authors and the comparison of the surface roughness behavior of dental composites in the literature will enrich the study. (for example Effect of thermal cycling and microhardness on roughness of composite restorative materials, ) In the literature, it has been reported that a smooth surface, which composite material, both improve the aesthetic properties of material and reduces plaque retention. (1-Characterization of water sorption, solubility, and roughness of silorane and methacrylate based composite resins. and 2- Investigation of two-body wear resistance of composite materials for biomaterial application in oral environment: the influence of antagonist material. and 3- Investigation of three‐body wear behavior and hardness of experimental titanium alloys for dental applications in oral environment
2) The authors should mention the particle size in the composite material (especially the nano and micro structure limits) and if they can visualize this structure with the shape, the reader will understand it more easily. In addition, the authors should also mention the aging processes of composite materials. Because the aging process is an inevitable environment structure that dentak composite materials are exposed to. (1- Investigating the effect of chewing force and an abrasive medium on the wear resistance of composite materials by chewing simulation)
3) Authors should provide literature information on wear mechanisms (for example two-body, three-body, fatigue wear etc). In this way, the study will turn into a more qualified structure ( you can see 1-A Literature Review on Chewing Simulation and Wear Mechanisms of Dental Biomaterials, 2- Investigation of two-body wear behavior of zirconia-reinforced lithium silicate glass-ceramic for biomedical applications; in vitro chewing simulation. 1_ Mesquita RV, Geis-Gerstorfer J. Influence of temperature on the visco-elastic properties of direct and indirect dental composite resins. Because the chewing process has a complex and continuous structure. With these additions, scientists studying this work will be able to better analyze the structure of this process. The work is written with a few spelling mistakes. Authors need to correct the mistakes with examining the publication from start to finish.
Recommendation: Manuscript can be accepted after minor revision

Author Response
1) The authors have added a paragraph on this topic and the references of the following suggested papers:
Yilmaz, E.Ç. Investigation of three-body wear behavior and hardness of experimental titanium alloys for dental applications in oral environment. 2020. Materialwiss. Werkstofftech. 51, 47.
Yilmaz, E.Ç. Investigation of two-body wear resistance of composite materials for biomaterial application in oral environment: the influence of antagonist material. 2020 Materials Technology, 35:3, 159-167, DOI: 10.1080/10667857.2019.1660476
Yilmaz, E.Ç.Effect of thermal cycling and microhardness on roughness of composite restorative materials. 2016. Journal of Restorative Dentistry 4(3):93
2) The authors have added more filler particle size information in the materials' compositon table. The authors also referenced to the aging process and added the following suggested reference:
Yilmaz, E.Ç. Investigating the Effect of Chewing Force and an Abrasive Medium on the Wear Resistance of Composite Materials by Chewing Simulation. Mech Compos Mater 56, 261–268 (2020). https://doi.org/10.1007/s11029-020-09878-2
3) The authors provided literature information on wear mechanisms adding two paragraphs and the following references:
Yilmaz, E.Ç., Sadeler, R. A Literature Review on Chewing Simulation and Wear Mechanisms of Dental Biomaterials. J Bio Tribo Corros 7, 91 (2021). https://doi.org/10.1007/s40735-021-00529-0
Yilmaz EÇ. Investigation of two-body wear behavior of zirconia-reinforced lithium silicate glass-ceramic for biomedical applications; in vitro chewing simulation [published online ahead of print, 2020 Nov 30]. Comput Methods Biomech Biomed Engin. 2020;1-19. doi:10.1080/10255842.2020.1852555
Reviewer 2 Report
The work is interesting, but it would be necessary to do another reading to correct spelling errors and if possible, add a real SEM image for the calculation used.
Author Response
Thank you for the comment. Figure 4, showing a real SEM image used for calculation, was added.
Reviewer 3 Report
1. The abstract requires the addition of quantitative results.
2. Please end your abstract with a "take-home" message.
3. Keywords should be reordered based on alphabetical order.
4. I think it is better for do not to use abbreviations in the keywords section.
5. In terms of biomaterials used, the authors encouraged to discuss it more comprehensive in perspective of biocompability, biomechanics, and biomimetic. For support the explanation. Please refer the relevant reference as follows: Minimizing Risk of Failure from Ceramic-on-Ceramic Total Hip Prosthesis by Selecting Ceramic Materials Based on Tresca Stress. Sustainability 2022, 14, 13413. https://doi.org/10.3390/su142013413
6. In the present form, actually nothing really novel. The current works appears to be a replication or modified literature according to the lack of novelty. The authors must extensively describe the novel in their work. This work should be rejected due to a serious concern.
7. The work, novelty, and constraints of relevant previous literature must be explained in the introduction section to highlight the research gaps that the present study aims to fill.
8. Please make the last paragraph of the introduction section describe the present article objective more clearly.
9. Additional figures in the introduction would improve the quality of the present article. Please provide it.
10. Recommended to the authors for providing workflow of the present work in form of figure to make it simple to understand rather than using dominant text to explain.
11. It's also important to provide more particular information on tools, such as the manufacturer, the country, and the specification.
12. The paper needs to provide critical information on the error and tolerance of the experimental equipment utilized in this work. Due to the disparate outcomes of subsequent research by other researchers, it would make for an insightful conversation.
13. Results comparison with similar previous studies needs to give.
Author Response
1) In the abstract the following quantitative results are listed:
The lowest marginal gap value was reported by zirconia (21.45±12.58mm), followed by composite (44.7±24.96mm) and lithium disilicate (62.28±51.8mm)
2) A clinical significance paragraph has been added to the abstract
3) Keywords have been reordered based on alphabetical order 4) SEM has been changed to "Scanning Electron Microscope". 5) In further studies the authors highlighted the need of discussing the biomaterials discussing them more comprehensive in perspective of biocompability,biomechanics, and biomimetic and cyting the relevant reference:
Ammarullah MI, Santoso G, Sugiharto S, Supriyono T, Wibowo DB, Kurdi O, Tauviqirrahman M, Jamari J. Minimizing Risk of Failure from Ceramic-on-Ceramic Total Hip Prosthesis by Selecting Ceramic Materials Based on Tresca Stress. Sustainability. 2022; 14(20):13413. https://doi.org/10.3390/su142013413
6) Two extensive paragraphs have been added to the discussion outlining the novelty and the clinical relevance of the findings of the current study.
7) In the introduction the conflicting results of scientific evidences have been outlined and therefore the need of the conducted study has been more clearly stated. The discussion of the results of the present study in respect to the available literature has been covered in the Discussion
8) The last paragraph has been improved.
The present article objective has been described more clearly.
9) Figure 4 (a 580x SEM image) for evaluation of marginal gap at restoration-abutment interface was added.
10) Figure 6 with the schematic representation of the experimental's design workflow was added.
11) Manufacturer, country and other missing information was added where missing. 12) Tolerance information (0.001mm) was added to the Materials and Methods section.13) Results were compared in the discussion with the following previous studies:
Gold, S.A.; Ferracane, J.L.; da Costa, J. Effect of Crystallization Firing on Marginal Gap of CAD/CAM Fabricated Lithium Disilicate Crowns. J Prosthodont 2018, 27, 63–66, doi:10.1111/jopr.12638.
Baig, M.R.; Al-Tarakemah, Y.; Kasim, N.H.A.; Omar, R. Evaluation of the Marginal Fit of a CAD/CAM Zirconia-Based Ceramic Crown System. The International Journal of Prosthodontics 2021, doi:10.11607/ijp.6654.
Alghazzawi, T.F.; Liu, P.-R.; Essig, M.E. The Effect of Different Fabrication Steps on the Marginal Adaptation of Two Types of Glass-Infiltrated Ceramic Crown Copings Fabricated by CAD/CAM Technology. Journal of Prosthodontics: Official Journal of the American College of Prosthodontists 2012, 21, 167–172, doi:10.1111/j.1532-849X.2011.00803.x.
Karataşli, O.; Kursoğlu, P.; Capa, N.; Kazazoğlu, E. Comparison of the Marginal Fit of Different Coping Materials and Designs Produced by Computer Aided Manufacturing Systems. Dental Materials Journal 2011, 30, 97–102.
Moldovan, O.; Luthardt, R.G.; Corcodel, N.; Rudolph, H. Three-Dimensional Fit of CAD/CAM-Made Zirconia Copings. Dental Materials: Official Publication of the Academy of Dental Materials 2011, 27, 1273–1278, doi:10.1016/j.dental.2011.09.006.
Anadioti, E.; Aquilino, S.A.; Gratton, D.G.; Holloway, J.A.; Denry, I.; Thomas, G.W.; Qian, F. 3D and 2D Marginal Fit of Pressed and CAD/CAM Lithium Disilicate Crowns Made from Digital and Conventional Impressions. J Prosthodont2014, 23, 610–617, doi:10.1111/jopr.12180.
Alqahtani, F. Marginal Fit of All-Ceramic Crowns Fabricated Using Two Extraoral CAD/CAM Systems in Comparison with the Conventional Technique. Clin Cosmet Investig Dent 2017, 9, 13–18, doi:10.2147/CCIDE.S130856.
Kim, J.-H.; Jeong, J.-H.; Lee, J.-H.; Cho, H.-W. Fit of Lithium Disilicate Crowns Fabricated from Conventional and Digital Impressions Assessed with Micro-CT. J Prosthet Dent 2016, 116, 551–557, doi:10.1016/j.prosdent.2016.03.028.
Mostafa, N.Z.; Ruse, N.D.; Ford, N.L.; Carvalho, R.M.; Wyatt, C.C.L. Marginal Fit of Lithium Disilicate Crowns Fabricated Using Conventional and Digital Methodology: A Three-Dimensional Analysis. J Prosthodont 2018, 27, 145–152, doi:10.1111/jopr.12656
Round 2
Reviewer 3 Report
1. Overall, the discussion in the present article is extremely poor. The Authors must extend their discussion and make a comprehensive explanation.
2. Line 32-34, giving additional reference to support the explanation of digital technique that allow to short time and cost. The reference as follows: Adopted Walking Condition for Computational Simulation Approach on Bearing of Hip Joint Prosthesis: Review over the Past 30 Years. Heliyon 2022, 8, e12050. https://doi.org/10.1016/j.heliyon.2022.e12050
3. I believe not all of the limitations have been mentioned in the present manuscript,
4. Establish the conclusion as a paragraph rather than in its existing form, which is point by point.
5. Further research should be discussed in the conclusion section.
6. In the whole text, the authors created paragraphs that were only one or two phrases long, making the explanation unclear. The authors should extend on their explanation to generate a more detailed paragraph. It is advised each paragraph contain at least three sentences, with one sentence providing as the primary sentence and the other sentences providing as supporting sentences.
7. The reference is recommended to be enriched with literature from five years ago. MDPI reference is strongly recommended.
8. The authors are encouraged to reduce their self-citation.
9. The authors need to proofread the manuscript due to grammatical errors and language style.
10. Following the revision step, the authors must provide a graphical abstract.
Author Response
- Discussion has been extended. Now is 2460 words long.
- This reference has been added.
- Limitation section has been extended.
- The conclusion has been expanded.
- Further needed research is outlined in line 392-395.
- The manuscript has been edited by an external service proofreading text and syntax. Paragraph issues has been addressed by a native english speaking service.
- The authors checked the references, considering the inserted references as relevant. There are 93 references that the authors consider a consistent amount of references.
Nevertheless, if the reviewer thinks that the authors should extend more the references, we can ask the academic editor to consider this matter. We could ask if further references (more than 93) are really needed and if some of the recently suggested references, that we added, that are not related to the dental field (hip joint) both produced by Jamari J, Ammarullah MI, Santoso G, et al. are really pertinent to this paper. - The authors checked the references, considering all the inserted references as relevant. See point 7.
- The manuscript has been edited by an external service proofreading text and syntax.
- Figure 6 shows a schematic representation of the experimental's design workflow.